# Lifestyle and Treatment Adherence Intervention after a Coronary Event Based on an Interactive Web Application (EVITE): Randomized Controlled Clinical Trial Protocol

**DOI:** 10.3390/nu13061818

**Published:** 2021-05-27

**Authors:** María Ángeles Bernal-Jiménez, Germán Calle-Pérez, Alejandro Gutiérrez-Barrios, Livia Gheorghe, Ana María Solano-Mulero, Amelia Rodríguez-Martín, Josep A. Tur, Rafael Vázquez-García, María José Santi-Cano

**Affiliations:** 1Faculty of Nursing and Physiotherapy, University of Cádiz, 11009 Cádiz, Spain; m.angeles.bernal@inibica.es (M.Á.B.-J.); amelia.rodriguez@uca.es (A.R.-M.); 2Institute of Biomedical Research and Innovation of Cádiz (INiBICA), 11009 Cádiz, Spain; gcallep@hotmail.com (G.C.-P.); aleklos@hotmail.com (A.G.-B.); livia_gheorghe_ro@yahoo.es (L.G.); rafael.vazquez.sspa@juntadeandalucia.es (R.V.-G.); 3Cardiology Unit, Puerta del Mar Hospital, 11009 Cadiz, Spain; anamasom@gmail.com; 4Biomedicine, Biotechnology and Public Health Department, University of Cadiz, 11003 Cádiz, Spain; 5Research Group on Community Nutrition & Oxidative Stress, University of the Balearic Islands, IDISBA & CIBEROBN, 07122 Palma de Mallorca, Spain; pep.tur@uib.es; 6Research Group on Nutrition, Molecular, Pathophysiological and Social Issues, University of Cádiz, 11009 Cádiz, Spain

**Keywords:** lifestyle, coronary event, online application, coronary heart disease, treatment adherence, secondary prevention, randomized controlled clinical trial (RCT)

## Abstract

Coronary heart disease is one of the main causes of morbimortality around the world. Patients that survive a coronary event suffer a high risk of readmission, relapse and mortality, attributed to the sub-optimal control of cardiovascular risk factors (CVRF), which highlights the need to improve secondary prevention strategies aimed at improving their lifestyle and adherence to treatment. Through a randomized controlled clinical trial, this study aims to evaluate the effect of an intervention involving an online health application supported by a mobile telephone or tablet (mHealth) on lifestyle (diet, physical activity, and tobacco consumption) and treatment adherence among people with coronary heart disease after percutaneous coronary intervention. The sample will comprise 240 subjects (120 in each arm: intervention and usual care). They are assessed immediately and nine months after their hospital discharge about sociodemographic, clinical, CVRF, lifestyle, and treatment adherence characteristics. The educative intervention, involving a follow-up and self-monitoring, will be performed using an online mHealth tool consisting of an application for mobile phones and tablets. The quantitative primary outcomes from the two groups will be compared using an analysis of covariance (ANCOVA) adjusted for age and gender. A multivariate analysis will be performed to examine the association of the intervention with lifestyle habits, the control of CVRFs, and outcomes after discharge in terms of the use of health services, emergency visits, cardiovascular events and readmissions.

## 1. Introduction

Coronary heart disease is one of the main causes of morbimortality around the world [1]. Although progress has been made in the treatment of cardiovascular risk factors (CVRF) such as hypolipidemic and antihypertensive drugs, the results have fallen short of expectations. This is partly a result of the increased prevalence of obesity and type-2 diabetes mellitus, although the ageing population is also partly to blame [2]. Coronary interventions with stenting have achieved excellent clinical outcomes among these patients. However, although this procedure restores blood flow to the heart, it does not protect the myocardium against the onset of further coronary obstructions [3,4].

Thus, a high risk of readmission, relapse, and mortality has been seen to exist in the years following a myocardial infarction due to inadequate control of cardiovascular risk factors, highlighting the need to improve secondary prevention strategies [5,6]. Consequently, the first step is to educate patients about lifestyle habits and how to control CVRFs, in addition to conducting cardiac rehabilitation and complying with suitable pharmacological treatment.

Secondary prevention strategies aimed at changing these lifestyle habits and changing CVRFs are cost-effective thanks to the savings in both medication and future catheterization interventions or surgery [7,8,9]. However, a large number of patients do not comply with lifestyle recommendations or do not take their medication appropriately after a cardiovascular event, which results in them not reaching their treatment objectives. Among these patients, 30% stop the treatment three months after the first heart attack, while 50% abandon it after a year [10,11]. For this reason, patients need help and information about making changes in their lifestyles and adhering to their pharmacological treatment. This advice should be provided by the medical team (doctors and nurses) in the hospital and given before patients are discharged, as recommended in clinical practice guidelines [10].

In turn, patients must have the knowledge, means, and institutional support to take care of their health. However, they often encounter difficulties understanding the information they receive because the time short allotted for appointments makes detailed explanations impossible, which has been recognized as one of the hurdles in understanding the risk they are facing [12].

It has been shown that intensive, structured interventions lead to positive changes in the lifestyle of patients with coronary heart disease. They achieve improvements in CVRFs, contribute to the prevention of events and reduce hospital admissions [10,13,14,15,16], even being more successful in secondary than in primary prevention [12,17].

The development of new online communication channels between health professionals and patients may help to improve the care given by the latter. This technology also makes patients more self-reliant in the monitoring of their illness, allowing them to play a more active role while enabling health professionals to obtain more information about the health status of their patients.

Currently, over 75% of the world’s population has a mobile telephone with internet access, and over 57% of homes have an internet connection. In Europe, those figures reach 99% and 86% respectively [18,19].

These data suggest that smartphones have become a part of many aspects of day-to-day life and that they are potentially useful tools for managing health and lifestyle, and can be incorporated into communication between health professionals and patients.

Accordingly, the American Health Association issued a scientific statement in 2015 urging for an increase in research to assess the viability and efficacy of mobile health applications in the prevention of cardiovascular disease (CVD) [20].

Several clinical trials with varying intensity, duration, interventional components and contexts have analyzed the efficacy of using mobile devices as a tool for enhancing health-promoting habits and encouraging adherence to treatment in different populations. The mActive clinical trial evaluated the capacity of a system of automated and personalized text messages to increase physical activity, reporting a 25% increase in physical activity among the study group and concluding that text messaging acted as a driver of lifestyle changes. However, the limitations of this study were its short duration and the limited sample size [21].

In turn, the TEXT ME clinical trial studied the use of sending text messages to patients with coronary heart disease over six months, and found improvements in the intervention group regarding the CVRF objectives recommended in guidelines [22]. Other studies have also attempted to determine the effect of text messages on adherence to medication among coronary heart disease patients, but with differing results and limited effect sizes [23,24].

More recent clinical trials have studied the use of mobile phone applications as potential tools for improving adherence to treatment, the control of CVRFs, and the clinical outcomes of patients that suffered from a coronary event [25,26,27]. These patients are generally overwhelmed by a series of complex prescriptions for medicines, diets, and other advice.

In this sense, several systematic reviews and meta-analyses have analyzed the effectiveness of mHealth interventions in improving treatment adherence and self-monitoring of cardiovascular risk factors in patients with heart disease, although disparate results were found [28,29,30]. However, although a range of behaviors can be modified using mobile applications, there is a need for more studies of longer duration to provide evidence about the efficacy. These should better describe the characteristics of the intervention and the patients to fill the existing knowledge gap. In addition to extending the study for nine months, the present project involves a structured, systematic methodology that will enable the results to be compared with other international studies. [31].

This study aims to evaluate, through use of a randomized controlled clinical trial (RCT), the effect of an intervention involving a mobile health application on the lifestyle and treatment adherence of people with coronary heart disease after percutaneous coronary intervention in terms of:Improvements in the pattern of diet and food consumption, physical activity and smoking habits.The objective measures of cardiovascular risk such as body mass index (BMI), waist circumference, blood pressure, and low-density lipoprotein (LDL) cholesterol.Acquiring knowledge about cardiovascular diseases (CVD), the risk factors, and a healthy lifestyle.The patients’ level of anxiety, quality of life, and well-being.The participants’ commitment to using the application and the level of self-monitoring of CVRFs.The participants’ satisfaction regarding the usability of the application.Use of health services, emergency visits, and readmissions of the participants throughout the study.

## 2. Materials and Methods

### 2.1. Design

A randomized single-blind, parallel-group, controlled clinical trial performed on patients with coronary heart disease who underwent a percutaneous coronary intervention (PCI) with stent placement in the Cardiology Service of a public reference hospital complex providing specialist care in the province of Cadiz, Spain, in which 1500 coronary interventions procedures are performed per year (Figure 1).

### 2.2. Randomization and Blinding

The randomization and allocation to each group (1:1, intervention and usual care) are based on computer-generated random numbers. The researchers responsible for the study do not participate in the allocation of the participants. Due to the kind of intervention, blinding is not possible when the participants are allocated to groups. To minimize any bias, objective clinical variables are measured in the evaluation visit and the analyses are performed by blinded researchers.

### 2.3. Study Sample

The participants are eligible to participate if they have a confirmed diagnosis of coronary heart disease and undergo stenting with PCI. Furthermore, the participants must comply with the following criteria.

#### 2.3.1. Inclusion Criteria

Adults over 18 and below 75 years of age who have a smartphone that can receive messages and connect to the internet throughout the study period.

#### 2.3.2. Exclusion Criteria

Severe heart failure, physical disability or dementia and severe congenital, structural or rheumatic heart disease, chronic kidney or liver disease, if they are already using a health-monitoring application, or the application is not compatible with their telephone.

### 2.4. Sample Size

To detect a medium effect size of Cohen’s d of 0.5 [32] regarding adherence to the Mediterranean diet (8.6 ± 2.0 puntos) [33], food consumption, vegetables (471.4 g/day ± 230.0 g/day), fruit (308.4 g/day ± 188.6 g/day), meat and derived products (149.7 g/day ± 63.7 g/day), fish (122.3 g/day ± 73.5 g/day) [34], physical activity (210.2 METs-min/day ± 221.8 METs-min/day) [35,36] and a 12% decrease in smoking habits (prevalence of 21% in pilot study), a 95% confidence interval and a power of 90%, the sample size is estimated at 100 patients in each group. Assuming a 20% loss to follow-up with 240 participants, 120 in each arm: intervention and usual care.

### 2.5. Recruitment

After the PCI and during admission, the nurse will recruit eligible patients and their care partners, will ask them to sign the informed consent, perform the initial assessment and organize a programmed visit after 36 weeks. A card will be provided with the date of the appointment and a telephone number for any changes.

The participants allocated to the usual care group receive the standard prescribed care and advice about medication, and lifestyle.

Both groups will be provided with written recommendations and an explanation about the standard Mediterranean diet, physical activity, stopping smoking and treatment adherence.

Before hospital discharge, all the patients will be encouraged to follow a healthy lifestyle. Stages of change strategies will be used in addition to a motivational and behavior changing interview [37,38]. Written information will be provided about risk factors, lifestyle goals, a suggested healthy menu, recommendations about the daily intake of food groups [39], and the other behavior that the intervention is targeting.

Currently, this study has recruited 95 patients.

### 2.6. Intervention

The intervention begins during the patient’s stay in hospital immediately after a coronary event. The participants from the intervention group and their partner/carer will complete a short online tutorial describing the mobile application accessed using a mobile telephone or tablet (Figure 2). They will be advised to use the application for at least 15 min per day. This time has been considered sufficient for the daily recording of data in the pilot study. The intervention will last 36 weeks. If the patient does not record data for a week, he/she receives a message through the app encouraging him to use it. The patients may resolve any queries using the application’s built-in messaging function, to which the nurse will reply through this messaging service or with a telephone call. This avoids many patient visits to the doctor for consultations and reduces human resource needs.

Before the trial, a pilot study was performed with 20 participants to test the application and make any necessary adjustments.

### 2.7. Technical Data of the Website and Application

The responsive online application (user registration, data management, downloading records) is managed via the project website, which acts as an access platform after validation with a username and password. The web environment enables the application to be executed using any browser. Operating environment: it is an application with remote access to a MySQL database. Technology development: (a) uses PHP scripting language (Personal Home Page Tools); (b) AJAX web development techniques (Asynchronous JavaScript and XML). The application runs in the user’s browser while it communicates with the server in the background; and (c) Bootstrap open-source tools for designing websites and online applications. Data storage: MySQL database is fast enough to deploy web applications. Data protection: in addition to on-demand backups performed by the software, the web server performs daily backups of all the files, so the data and program are doubly protected. Access privacy: The data are not stored in a local computer but on a web server, meaning they can only be accessed with a username and password. This web server works with anonymous data and is located in the country to comply with the regulations for the protection of high-level data.

### 2.8. Application Contents

The application allows users to set goals and monitor their food consumption, physical exercise, blood pressure, tobacco use, and compliance with their treatment. It is based on the phases of change theory (attention, retention, memory, action, and motivation) [37] and on making the process pleasing [40,41]. The user’s attention is caught through warnings and bright, attractive colors on the user interface; retention is encouraged by reminders, repetition and graphs; action is prompted by instructions, advice, and feedback; and motivation to change is boosted by internal comparisons (progress graphs), setting goals, self-monitoring and feedback.

Through its different components (website, messages, emails, and calls), participants are encouraged to (1) follow a healthy eating pattern based on the Mediterranean diet aligned with national dietary guidelines [39]; (2) perform physical activity of duration and intensity in line with the recommendations of their cardiologist; (3) stop smoking; (4) monitor their blood pressure; (5) adhere to their treatment by associating taking medication with daily activities, establishing set times for taking it and with support from a relative, etc.

### 2.9. Components of the Application

A. Provide information encouraging a healthy lifestyle. Through the website the participants will have access to information on their screens that they can print to help them plan a healthier lifestyle and adhere to their treatment. The application also has a training section for the patient with information about the recommended therapeutic objectives in the clinical practice guidelines regarding food, physical activity, body weight, blood pressure, blood sugar, stopping smoking and adhering to treatment.

B. Self-monitoring. The application has a recording and self-checking function to help patients to self-monitor the skills for each behavioral goal about nutrition, physical activity, tobacco consumption, blood pressure, body weight, capillary blood glucose in patients with diabetes mellitus and treatment adherence.


Nutrition (Figure 3). The food eaten in each meal (breakfast, lunch, tea, and dinner) is recorded using a drop-down list from which the food, and the amount consumed are selected (a portion, half portion or quarter portion). This is preferably recorded daily, or at least once a week.Physical activity and rest (Figure 3). The patient can select from a drop-down list the type of activity performed during the day, the duration, and the total number of steps each day. Additionally, they record the number of hours of sleep from the previous night and the minutes they are seated or having a siesta.Treatment (Figure 3). The participant accesses a screen with a personalized list with their daily treatment (name of medication, dose and timetable) and indicates the medication is taken, which is recorded. Medication cannot be recorded in advance.



Body weight (Figure 4). The participant records their body weight and waist circumference every week. The application calculates their BMI and classifies the value as normal, overweight, or obese, following the WHO BMI classification for adults [42]. The size in cm is recorded at the beginning when the application is activated.Blood pressure (Figure 4). The participants record their systolic and diastolic blood pressure as well as their heart rate every week.Blood glucose in patients with diabetes mellitus (Figure 4). Twice a week, the patients record their fasting blood glucose and levels two hours after consuming food.Tobacco in smokers (Figure 4). Record the number of cigarettes smoked in a week.


C. Motivate patients to improve and maintain lifestyle habits. 1. By automatic text message reminders about healthy habits generated randomly on a pop-up screen (once a week). 2. By personalized messages about reaching goals related to improving their lifestyle and treatment adherence, and recommendations about aspects to be improved. (Figure 5). The messages are produced in response to the information recorded by the patient over the previous seven days. The patients, therefore, receive weekly feedback via pop-ups that appear when the application is opened on Mondays and Tuesdays (after reading the information on the pop-up screen, it can be closed). The messages appear as short sentences in green, yellow, and red depending on the degree of compliance and control of the goals. The message is in green when the set goals have been reached; in yellow if they have partially reached; and in red when the goal is pending (Figure 5).

As a reminder of achievements and aspects to improve, on the other days of the week (Wednesday to Sunday) the top left corner of each section of the main screen of the application appears in the color corresponding to the goals reached during the previous week. (Figure 5).

### 2.10. Training Session about the Application for Patient and Carer

The nurse will install the shortcut to the application on the screen of the participant’s mobile phone and will record the prescribed treatment including the name, dose, and timetable in the application. The nurse will update the prescription in the application if the doctor changes the treatment. During the training session, the participants learn to use the functions of the application: confirm when medication is taken, record the food consumed and physical activity performed (daily), blood pressure, weight, and tobacco consumption (weekly), and capillary blood glucose in participants with diabetes mellitus (twice a week).

The participants can follow their evolution and progress through the graphics generated with the information they have recorded over the previous eight weeks (Figure 6).

### 2.11. Ethical Considerations

The study will be conducted in agreement with the guidelines and protocols established in the Helsinki Declaration as revised in Fortaleza (Brazil) in October 2013, and complies with Law 14/2007 on Biomedical Research and with European Data Protection Regulations. It was approved by the Biomedical Research Ethics Committee of the Costa del Sol, Andalusia, with the reference: 003_ene19_PI-EVITE-18. The informed written consent of all the patients will be requested. The application guarantees the security measures regarding the current General Data Protection Regulations in Europe. It also includes data encryption mechanisms.

Clinical trial number in Trial: NCT04118504.

## 3. Results

Thirty-six weeks after the initial assessment, all the patients are reassessed in terms of their lifestyle, risk factors, and treatment management. In the intervention group, the variables related to usability, and satisfaction with the application will also be appraised. The participant’s satisfaction with the treatment received will also be evaluated (using a questionnaire). Records will be taken of whether the patient has been referred to cardiac rehabilitation and their attendance. Records will also be kept of the cardiovascular events occurring during the study, the use of health services and emergency visits taken from electronic medical records.

### 3.1. Variables Collected in the Initial and Final Evaluations

All the participants are evaluated at the beginning and end of the study though an interview, checking the information given against their electronic medical records. During hospitalization, after the PCI, an assessment is performed of sociodemographic characteristics, educational level, alcohol intake, tobacco consumption, and clinical situation (history of diabetes mellitus and its complications, high blood pressure, hyperlipidemia, previous cardiovascular events, number of stents implanted, Euroscore II scale classification of the left ventricular ejection fraction (LVEF)) [43]. Body weight, height (to calculate BMI), waist circumference [42], and systolic and diastolic blood pressure (mm Hg) will be measured. An examination will be performed of plasma concentrations of LDL cholesterol (mg/dL), high-density lipoprotein (HDL) cholesterol (mg/dL), and baseline blood glucose and glycosylated hemoglobin (HbA1c) in patients with diabetes corresponding to levels in analyses performed before the intervention and after the final evaluation. Tobacco consumption will be assessed by the self-reported number of cigarettes per day and the Fagerstrom nicotine dependence test. [44,45] Stopping smoking will be self-reported and confirmed by a concentration of carbon monoxide in exhaled air of fewer than six parts per million measured with a co-oximeter CO CHECK PRO (MD Diagnostics LTD) at the beginning and end of the study. To evaluate dietary intake patterns, validated questionnaires will be used for assessing adherence to the Mediterranean diet and the frequency with which foods are consumed [33,34]. To convert the food consumed into kilocalories and nutrients, Spanish food composition tables will be used [46]. Physical activity will be evaluated using the validated Spanish version of the Minnesota physical activity questionnaire (MET min/week) [35,36]. The time (minutes/day) for sedentary activity and sleep will also be recorded. Treatment adherence will be evaluated using the Morisky–Green test [47]. Knowledge of coronary disease, CVRFs, and a healthy lifestyle will be examined using a validated questionnaire [48]. An OMS questionnaire will be used to study levels of anxiety and depression [49], quality of life [50], and well-being [51].

Usability will be evaluated in the intervention group using a 21-item questionnaire completed after the intervention to assess the user’s acceptance of mHealth inventions [52]. Satisfaction with the application will be assessed using a specific questionnaire developed by the research team.

### 3.2. Variables Recorded during the Intervention Using the Online Application

During the study period, and only in the intervention group, variables will be examined corresponding to (1) dietary intake recorded for 24 h, 7 days/week (or at least once a week). The participants select an icon corresponding to the kind of food, portions and size of the portion that they have consumed during that day; (2) minutes and type of physical activity performed, and time spent on sedentary activities and sleep. Number of steps/day using an Onwalk 100 pedometer and accelerometer (Geonaute, Decathlon, Spain), 7 days/week; (3) tobacco consumption: mean number of cigarettes/day once a week; (4) treatment compliance: treatment taken seven days/week; (5) self-measured body weight once a week to calculate BMI; (6) systolic and diastolic blood pressure measured once a week; (7) fasting and postprandial capillary glucose in patients with insulin-dependent diabetes mellitus, two times per week; and (8) frequency of use of the application (number of records per day).

#### 3.2.1. Primary Outcome Variables

The primary outcome measures after nine months in both groups will be changes in behavior relating to 1. Healthy diet: Adherence to the Mediterranean diet and average intake of each food group. 2. Level of physical activity (MET) and the number of steps per day, and a less sedentary lifestyle. 3. Smokers stop smoking. 4. Treatment Adherence. 5. Knowledge acquired. 6. Usability and satisfaction with the application.

#### 3.2.2. Secondary Outcome Variables

The secondary outcome variables will be 1. BMI. 2. Waist circumference. 3. Blood pressure. 4. Total cholesterol and LDL cholesterol. 5. Capillary blood glucose, and blood glucose, and HbA1c in the patients with diabetes. 6. Cardiovascular events, use of health services, emergency visits, and readmissions throughout of the study.

### 3.3. Statistical Analysis

The principles of intention-to-treat analysis will be followed. The differences between patients who gave up the app and those who were able to utilize it will also be examined by analyzing the outcome variables from the electronic medical record. A descriptive statistical analysis will be performed (mean, standard deviation, 95% confidence interval, frequencies, and percentages). For the comparison of means, the Student’s t-test will be used (normal distribution variables) and the Mann–Whitney U test (if the variables are non-normally distributed). The Chi-squared/Fisher test will be used for the comparison of proportions. The relative risks and 95% confidence intervals will be computed. The quantitative primary outcomes from the two groups will be compared using an analysis of covariance adjusted for age and gender. A sensitivity analysis will also be performed regarding the baseline characteristics of the participants in both groups and missing data. A multivariate analysis will be performed to examine the association of the intervention with lifestyle habits, control of the CVRFs and evolution after discharge regarding cardiovascular events, the use of health services, emergency visits, and readmissions using Kaplan–Meier survival analysis. A two-tailed *p*-value < 0.05 will be considered statistically significant. The SPSS v.24.0 software will be used. 24.0. The researchers analyzing the results will be blinded to the allocation of the participants.

## 4. Discussion

Maintaining lifestyle changes and healthy behavior is the key to achieving improved clinical outcomes in patients with cardiovascular diseases (CVD). Thus, ongoing contact with the patient is necessary. The strategies used in studies to achieve behavioral changes and healthier lifestyles include: individual guidance, informing the patients of risks, shared decision-making, participation of family members, setting goals for treatment adherence, the control of CVRFs, and motivating participants. Modern technology can help in implementing these strategies. To this end, there has been increasing interest recently in research into mHealth resources for the prevention of CVD [21,27]. However, there is a lack of evidence about their efficacy and about the features of applications that increase commitment and effectiveness.

The present project acts to improve both lifestyle and treatment adherence through a remote, personalized intervention using limited material and human resources aimed at improving the outcome of these patients. The intervention is designed to be integrated into the care given to cardiac patients from the moment of their admission and aims to consolidate the activities the usual care given to these patients. Additionally, secondary prevention begins during the hospital stay, a stage in which the patients and their families are very receptive to advice from medical staff as the coronary event is very recent.

This study, involving a CRT, will provide fresh evidence about the viability and efficacy of a mHealth application for improving adherence to treatment and lifestyle among patients with coronary heart disease. An application will be tested that has been designed by health professionals and IT engineers following international guidelines [31]. In addition to the data recorded by patients in the application, objective clinical measurements will be taken such as BMI, waist circumference, blood pressure (BP), carbon monoxide in exhaled air, as well as laboratory results such as cholesterol, lipoproteins, blood glucose, and HbA1c. An analysis will also be performed of the usability of the application and the patients’ satisfaction and commitment, as well as important features of the application that can enhance its efficacy and use.

### Limitations of the Study

To use this tool, it is necessary to have a mobile telephone and internet connection, conditions that are met by a growing number of people of the age of the target population in our country, evidence of which is the current statistics presented in the introduction. The recruitment will take place in a city hospital and differences may exist in rural areas. However, the hospital treats patients from different regions of the province, thus providing information about the reach of this intervention. The possible participants are volunteers and are likely, therefore, to be more motivated, although this is a more general limitation regarding less motivated people and not a selection bias since the study design is a randomized clinical trial.

## 5. Conclusions

The use of new technology and applications for smartphones is an appealing tool in secondary prevention strategies for CVD. Therefore, there is a need for more evidence about their usefulness for monitoring patients that will be of use to health professionals and medical services. The study will provide evidence about the effectiveness of an mHealth intervention that may be transferable and used to encourage adherence to recommendations about healthy lifestyles and treatment with pharmacological treatment in cardiovascular disease and other chronic illnesses.

## Figures and Tables

**Figure 1 nutrients-13-01818-f001:**
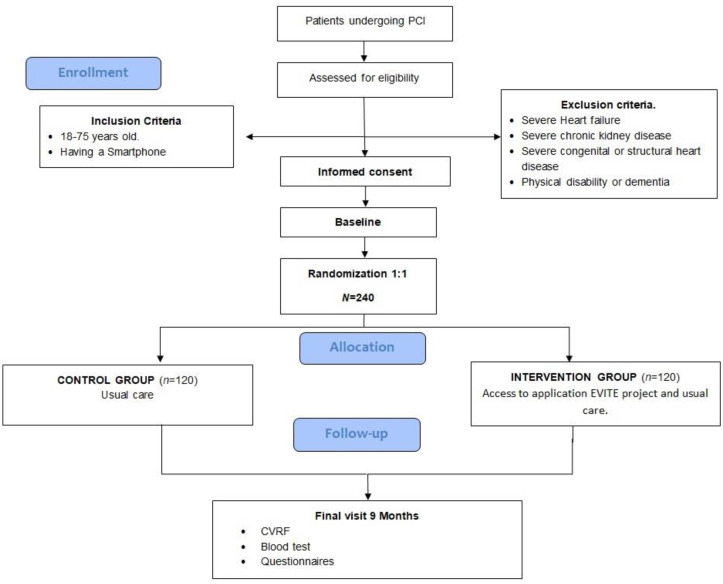
Flow Diagram. CVRF: cardiovascular Risk factors; PCI: percutaneous coronary intervention.

**Figure 2 nutrients-13-01818-f002:**
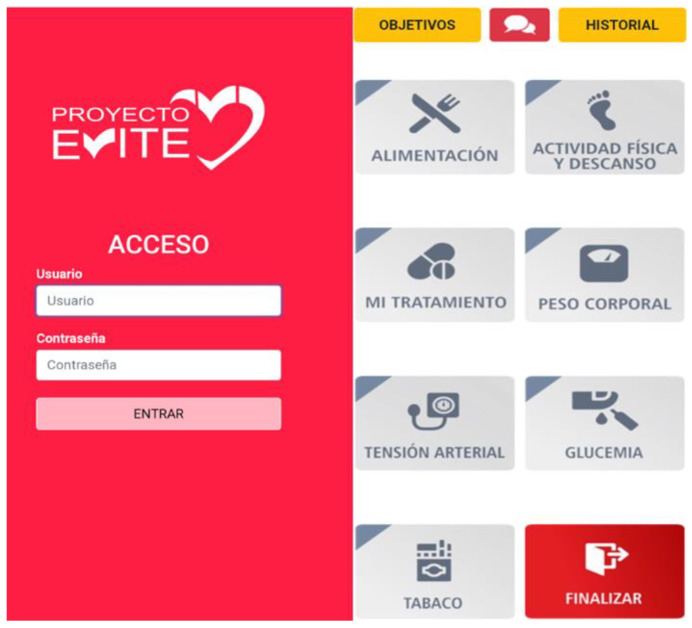
Start page app of the EVITE project.

**Figure 3 nutrients-13-01818-f003:**
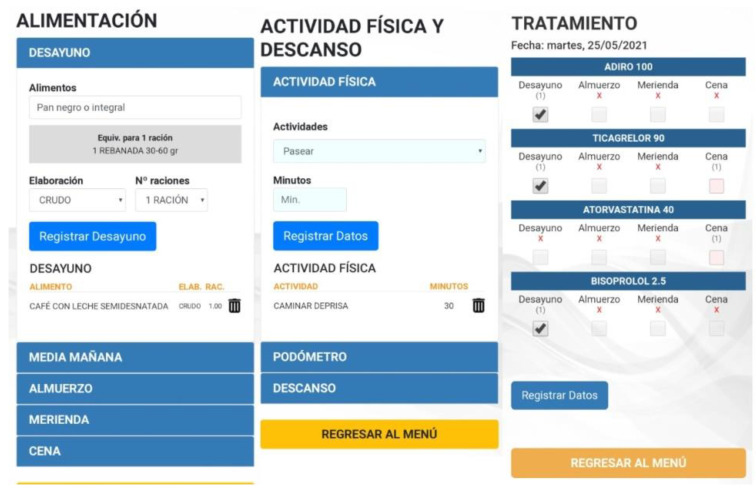
Diet, physical activity, and treatment modules (from left to right).

**Figure 4 nutrients-13-01818-f004:**
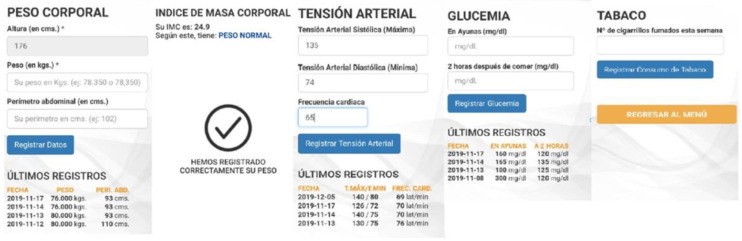
Body weight, body mass index (BMI), blood pressure, capillary blood glucose and tobacco modules (from left to right). * Body weight and height to calculate BMI.

**Figure 5 nutrients-13-01818-f005:**
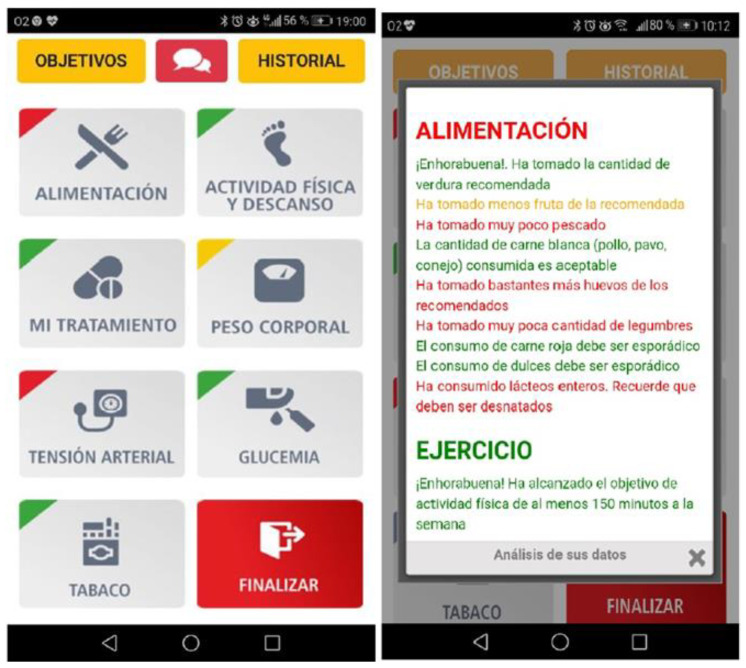
Weekly feedback traffic light and messages.

**Figure 6 nutrients-13-01818-f006:**
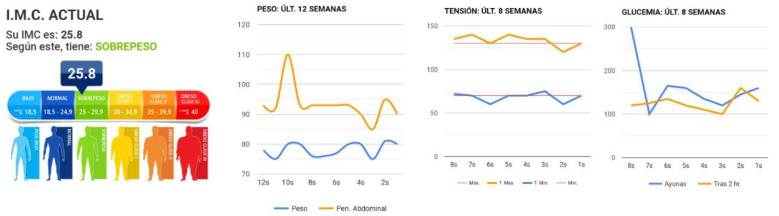
Record: graphics of evolution and progress of body mass index, blood pressure, body weight and waist circumference, and capillary blood glucose (from left to right).

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
