# Peer review of "Lifestyle and Treatment Adherence Intervention after a Coronary Event Based on an Interactive Web Application (EVITE): Randomized Controlled Clinical Trial Protocol"

_nutrients, 2021, doi:10.3390/nu13061818_

Round 1

Reviewer 1 Report

This study is a challenge to the secondary prevention of coronary artery disease, which has a high clinical need. The study design is well sophisticated and highly worth examining. The reviewer would like to ask for clarification on a few points

  1. Please clarify which and how many sites are eligible for the present study.
  2. Are there human interventions that encourage the use of the app? If not, it should be emphasized more, as the lack of need to invest human resources is a great advantage.
  3. How many minutes per day would be needed if a patient actually used the app completely?
  4. How well you can motivate patients by using the app will be the key to this study. The reviewer thought it would be good to examine the differences between patients who gave up the app and those who were able to utilize it.

Author Response

                                                                                       May 22, 2021.

Reviewer #1:

Thank you for reading our manuscript titled “Lifestyle and treatment adherence intervention after a coronary event based on an interactive web application (EVITE): Randomized controlled clinical trial protocol” (Manuscript ID: nutrients-1232505) and reviewing it, which has helped us improve it. We revised the manuscript and all suggested changes have taken place.

Please, find below our point-by-point answers to reviewer’s comments (blue color font). Changes made in the revised manuscript are in red letters.

 “This study is a challenge to the secondary prevention of coronary artery disease, which has a high clinical need. The study design is well sophisticated and highly worth examining. The reviewer would like to ask for clarification on a few points”

We thank the reviewer for these comments.

1. “Please clarify which and how many sites are eligible for the present study”.

In the revised manuscript, we have clarified in the Materials and Methods section (Page 3) this information:

 The study is carried out in the Cardiology Service of a public reference hospital complex providing specialist care in the province of Cadiz, Spain, in which around 1500 coronary intervention procedures are performed per year.

2. “Are there human interventions that encourage the use of the app? If not, it should be emphasized more, as the lack of need to invest human resources is a great advantage”.

In the revised manuscript, we have specified in the Materials and Methods section (Page 5) this information:

If the patient does not record data for a week, the patient receive a message through the app encouraging him to use it. The patients may resolve any queries using the application’s built-in messaging function, to which the nurse will reply through this messaging service or with a telephone call. This avoids many patient visits to the doctor for questions and reduces human resource needs.

3. “How many minutes per day would be needed if a patient actually used the app completely?”

 In the revised manuscript, we have specified in the Materials and Methods section (Page 5) this information:

They will be advised to use the application for at least 15 minutes per day. This time has been considered sufficient for the daily recording of data in the pilot study.

4. “How well you can motivate patients by using the app will be the key to this study. The reviewer thought it would be good to examine the differences between patients who gave up the app and those who were able to utilize it”.

In the revised manuscript, we have specified in the Materials and Methods section (Page 10) this information:

The differences between patients who gave up the app and those who were able to utilize it will be examined by analyzing the outcome variables from the electronic medical record.

We hope that the manuscript is now acceptable for publication in Nutrients.

Yours sincerely,

Mª José Santi, MD, PhD.

Reviewer 2 Report

The article by Bernal-Jiménez et al presents a randomized controlled clinical trial protocol aiming at the evaluation of effect of intervention involving an online health application supported by a smartphone or tablet (mHealth) on the treatment adherence and lifestyle in a population of 240 patients with coronary heart disease after percutaneous coronary intervention.

The article describes the step-by-step protocol that is going to take place, from enrolment and randomization of patients to training for the app utilization and the follow-up procedures. It is written in a clear and comprehensible manner. 

Author Response

                                                                                                 May 22, 2021.

Reviewer #2:

Thank you for reading our manuscript titled “Lifestyle and treatment adherence intervention after a coronary event based on an interactive web application (EVITE): Randomized controlled clinical trial protocol” (Manuscript ID: nutrients-1232505) and reviewing it.

 “The article by Bernal-Jiménez et al presents a randomized controlled clinical trial protocol aiming at the evaluation of effect of intervention involving an online health application supported by a smartphone or tablet (mHealth) on the treatment adherence and lifestyle in a population of 240 patients with coronary heart disease after percutaneous coronary intervention.

The article describes the step-by-step protocol that is going to take place, from enrolment and randomization of patients to training for the app utilization and the follow-up procedures. It is written in a clear and comprehensible manner”. 

We thank the reviewer for these comments.

Yours sincerely,

Mª José Santi, MD, PhD.
